# Exploring the achievements and forecasting of SDG 3 using machine learning algorithms: Bangladesh perspective

**Md. Maeen Molla**[1]*, **Md. Sifat Hossain**[2], **Md. Ayub Ali**[3], **Md. Raqibul Islam**[2], **Mst. Papia Sultana**[2], **Dulal Chandra Roy**[2]

**1** Department of Statistics, Pirojpur Science and Technology University, Pirojpur, Bangladesh, **2** Department of Statistics, University of Rajshahi, Rajshahi, Bangladesh, **3** Department of Statistics, University of Barishal, Barishal, Bangladesh

* s1710624163@ru.ac.bd

## Abstract

### Background

Sustainable Development Goal 3 (SDG 3), focusing on ensuring healthy lives and well-being for all, holds global significance and is particularly vital for Bangladesh. Neonatal Mortality Rate (NMR), Under-5 Mortality Rate (U5MR), Maternal Mortality Ratio (MMR) and Death Rate Due to Road Traffic Injuries (RTI) are considered responsible indicators of SDG 3 progress in Bangladesh. The objective of the study is to forecast these indicators of Bangladesh up to 2030 and compare these forecasts with predetermined 2030 targets. The data is obtained from the World Bank's (WB) website.

### Method

For forecasting, time series models were employed, specifically Autoregressive Integrated Moving Average- ARIMA (0,2,1) with Akaike Information Criterion (AIC) 94.6 for NMR and ARIMA (2,1,2) with AIC 423.2 for U5MR, selected based on their lowest AIC values. Additionally, Machine Learning (ML) models, including Bidirectional Recurrent Neural Networks (BRNN) and Elastic Neural Networks (ENET), were employed for all the indicators.

### Results

ENET demonstrates superior performance compared to both BRNN and ARIMA in the context of NMR, achieving a Root Mean Absolute Error (RMAE) of 0.603446 and a Root Mean Square Error (RMSE) of 0.451162. Furthermore, when considering U5MR, MMR, and Death Rate Due to RTI, ENET consistently exhibits lower error metrics compared to the alternative models. Following the time series and ML analyses, a consistent trend emerges in the forecasted values for NMR and U5MR, which consistently fall below their respective 2030 targets. This promising finding suggests that Bangladesh is making significant progress toward meeting its 2030 targets for NMR and U5MR. However, in the cases of MMR and Death Rate Due to RTI, the forecasted values exceeded 2030 targets.

**Data availability statement:** All data are in the paper and/or Supporting Information files. All the used dataset are publicly available in WB data bank. The NMR data of Bangladesh from the year 1960 to 2020 is available at https://data.worldbank.org/indicator/SH.DYN. NMRT?locations=BD. The U5MR data of Bangladesh from the year 1960 to 2020 is available at https://data.worldbank.org/indicator/SH.DYN.MORT?locations=BD. The MMR data of Bangladesh from the year 2000 to 2020 is available at https://data.worldbank.org/indicator/SH.MMR.DTHS?locations=BD. The death due to RTI data of Bangladesh from the year 2000 to 2019 is available at https://data.worldbank.org/indicator/SH.STA.TRAF.P5?locations=BD.

**Funding:** The author(s) received no specific funding for this work.

**Competing interests:** We have no competing interest.

This indicates that Bangladesh faces challenges in meeting the 2030 targets for MMR and Death Rate Due to RTI.

## Conclusion

The analyses underscore the importance of SDG 3 in Bangladesh and its progress towards ensuring healthy lives and well-being for all. While there is optimism regarding NMR and U5MR, more focused efforts may be needed to address the challenges posed by MMR and Death Rate Due to RTI to align with the 2030 targets. This study contributes valuable insights into Bangladesh's journey toward sustainable development in the realm of health and well-being.

## Introduction

Sustainable development, as defined by the United Nations, involves responsibly using renewable resources to meet current needs without compromising future generation's ability to meet their own needs. The Sustainable Development Goals (SDGs) replaced the Millennium Development Goals (MDGs) and consist of 169 specific targets in a global effort to combat poverty and its associated challenges [1]. To ensure the achievement of Sustainable Development Goals in Bangladesh, a specific set of four indicators has been chosen under SDG 3. These indicators include NMR, U5MR, MMR, and Death Rate Due to RTI. The selection of these indicators has been done under the guidance of the SDG Working Committee of The Prime Minister's Office. All relevant ministries are actively involved in this process [1]. It would be helpful to understand why these indicators are critical for Bangladesh. These indicators are important because they reflect the overall health and well-being of the population, especially vulnerable groups like newborns, children under 5, and mothers. High neonatal, under-5, and maternal mortality rates indicate challenges in healthcare access, quality, and maternal health. Deaths from RTI highlight the need for improved road safety measures. By focusing on these indicators, Bangladesh can prioritize interventions and policies to improve healthcare and save lives. Promoting the well-being of all and ensuring a healthy life for everyone is essential for sustainable development. The Sustainable Development Goals, adopted by United Nations member states in 2015, strive to eliminate preventable deaths among newborns and children under the age of 5. This involves the goal of decreasing neonatal mortality to a maximum of 12 deaths per 1,000 live births and under-five mortality to no more than 25 deaths per 1,000 live births by the year 2030 [2]. Another objective of SDG 3 is to decrease the worldwide MMR to below 70 per 100,000 live births by the year 2030. While substantial progress has been made in reducing maternal deaths globally, maternal mortality remains unacceptably high in many developing countries [3]. RTI also contribute significantly to mortality and pose a major threat to public health worldwide. SDG 3 aims to halve the number of global deaths and injuries from road traffic accidents by 2020 instead of 2030 [4]. Bangladesh has made notable progress in achieving many of the health-related MDGs. The NMR declined from 43 per 1,000 live births in 2000 to 19 in 2019. U5MR decreased from 93 to 29 deaths per 1000 live births during the same period [5]. However, further efforts are still needed to meet the more ambitious SDG targets. MMR reduced from 323 in 2010 to 180 deaths per 100,000 live births in 2017 but remains high [6]. Death rate due to RTI increased from 9.6 per 100,000 population in 2010 to 12.8 in 2016 [7]. Accurate forecasts of future trends in these health indicators are crucial for effective policymaking and resource allocation to achieve the SDG 3 targets. This study aims to forecast key health indicators of Bangladesh up to 2030 using

time series and ML models to evaluate progress towards SDG 3. Neonatal mortality refers to the occurrence of infant deaths within the initial 28 days of life. The NMR is calculated as the number of such deaths per 1,000 live births [8]. The NMR data of Bangladesh from the year 1960 to 2020 is available at the WB's website are used for this research [9]. From the descriptive study of this data, we found that the minimum value of NMR of Bangladesh is 17.5 and the maximum value is 102.7 with first quartile, median and third quartile 35.9, 65.6 and 93.5 respectively. Additionally, we can include that there is no unusual observation in this data. The U5MR is characterized as the likelihood of a child dying from birth up to the age of five years. This encompasses neonatal, post-neonatal, and childhood mortality, which pertain to fatalities occurring in children under the age of five. U5MR is also expressed as the number of such deaths per 1,000 live births [10]. The U5MR data of Bangladesh from the year 1960 to 2020 is available at the WB's website are used for this research [11]. From the descriptive study of this data, the minimum value of U5MR of Bangladesh is 29.1 and the maximum value is 371.3 along with first quartile, median and third quartile 64.6, 146.1 and 224.2 respectively. Additionally, there is no unusual observation in this data. Maternal mortality is defined as the death of a pregnant woman or a woman within 42 days after delivery, resulting from causes related to or aggravated by the pregnancy or its management. This definition excludes deaths caused by accidental or incidental factors [12]. MMR is usually expressed as the number of maternal deaths per 100,000 live births [13]. The MMR data of Bangladesh from the year 2000 to 2020 is available at the WB's website are used for this research [14]. From the descriptive study of this data, the minimum value of MMR of Bangladesh is 163 and the maximum value is 434 with first quartile, median and third quartile 200, 258 and 343 respectively. Additionally, there is no unusual observation in this data. And death caused by road accident is considered as the death due to RTI. Death Rate Due to RTI is measured per 100,000 people [7]. The death due to RTI data of Bangladesh from the year 2000 to 2019 is available at the WB's website are used for this research [15]. From the descriptive study of this data, the minimum value of the death due to RTI of Bangladesh is 10.8 which is suspected as an outlier along with one more datapoint, the maximum value is 17.6 with first quartile, median and third quartile 15.4, 15.9 and 16.9 respectively.

The study leverages a diverse set of modelling techniques, including ARIMA and neural networks, to forecast and assess SDG 3 indicators. By incorporating advanced ML methods, this research expands upon previous analyses and provides a comprehensive view of the trajectory toward achieving these goals. Prima et al. (2022), predicts the prevalence of first-day neonatal mortality in Bangladesh using ML techniques, including Decision Tree, Random Forest, Support Vector Machine, and Logistic Regression [16]. M. Khan, N. Khan, O. Rahman et al. (2021) projects that the U5MR in Bangladesh will further reduce to 176 per 1000 livebirths by 2030 [17]. MMR forecasting in Bangladesh is an important aspect of policymaking. The MMR in Bangladesh has shown a decline in recent years, but it still remains high. The major causes of maternal deaths in Bangladesh include haemorrhage and eclampsia, with a significant number of deaths occurring during childbirth and within two days of birth [18,19]. One notable finding from previous research is that linear trend projections indicated challenges in meeting the MMR target based on historical reduction rates [20]. This study will delve deeper into the factors contributing to this challenge and explore whether ML models offer insights that can help address this issue. The death rate due to RTI in Bangladesh is a major concern. Studies have shown that the mortality rate related to RTI is gradually rising, with an estimated mortality rate of 15.3 per 100,000 population in 2019 [21,22]. Building upon the work of Wei et al. (2019), this study focuses not only on mortality rates but also on the broader spectrum of SDG 3 targets [23]. The application of neural network models for assessing progress is particularly promising, given their demonstrated superiority in forecasting

health indicators. The study draws attention to the global relevance of forecasting health indicators. As evidenced by the work of Adeyinka et al. (2020), ML models have outperformed traditional statistical techniques in predicting U5MR for Nigeria [24]. This underscores the importance of incorporating ML in global health assessments.

ARIMA, BRNN, and ENET were chosen for this study because they are well-suited for handling small sample sizes or limited data points [25]. ARIMA is specifically designed for forecasting equally spaced univariate time series data. BRNN, with its bidirectional structure, can effectively utilize information from both past and future states, which can be advantageous when working with a small sample size. ENET, as a regularization technique, can handle the multicollinearity problem and perform feature selection simultaneously, making it a good choice for models with limited data points. So far, there have been very few research studies on the SDG 3 indicator variables in Bangladesh. None of these studies have included all of the indicators together, and most of them have used classical time series analysis for forecasting. Our study aims to fill this research gap by forecasting all of the SDG 3 indicators of Bangladesh together using both classical time series analysis and ML methods. We will compare the accuracy of the different models and compare the forecasted values with the given target values for the year 2030, which marks the end of the SDG era.

## Materials and methods

A time series is generated from past data, representing a collection of observations on a variable at consecutive time points or across successive time periods. It is a sequence of realization process. A very well-known classical time series model, ARIMA is used for forecasting NMR and U5MR. Whereas BRNN and ENET ML models are used for forecasting all the indicators.

### ARIMA

ARIMA is a widely-used statistical method for time series forecasting that combines three components: autoregressive (AR), integrated (I), and moving average (MA). The AR part involves regressing the variable on its own lagged values, predicting future values based on past values. The I part involves differencing the raw observations to make the time series stationary, ensuring constant statistical properties like mean and variance over time. The MA part models the error term as a linear combination of past error terms. An ARIMA model is represented as ARIMA(p, d, q), where p is the number of lag observations, d is the number of differencing operations, and q is the size of the moving average window. Box and Jenkins' (1976) introduced ARIMA model which forecasts equally spaced univariate time series data [26].

### BRNN

BRNN connect two hidden layers that operate in completely opposite directions to produce a singular output. In this type of generative deep learning, the output layer can simultaneously gather information from both past (backward) and future (forward) states. Schuster and Paliwal introduced BRNNs in 1997, aiming to augment the volume of input data accessible to the network [27].

### ENET

In the realm of statistical modeling, specifically in the fitting of linear or logistic regression models, the ENET is a regularization technique that blends the penalties from the lasso and ridge methods in a linear combination such a way that it can handle the multicollinearity problem and feature selection at a same time [28]. Overall, these models were selected because

of their relevance and ability to handle the specific challenges posed by small sample sizes or limited data points.

### Cross validation and performance measures

The univariate time series is divided into two parts for calculating performance measures. The first section is known as the training series with 75% of the data, and the second as the test series with 25% of the data. The training series is used for modeling, while the test series is used to assess the model's forecasting ability. Four performance measures viz. AIC, $R^2$ RMSE, and RMAE were used to assess forecasting accuracy. The AIC is a statistical measure used to evaluate the quality of a model by balancing its goodness of fit and complexity. It is calculated as $AIC = 2k - 2ln(L)$, where k is the number of parameters in the model and $ln(L)$ is the log-likelihood of the model. A lower AIC value indicates a better model fit. $R^2$ is a statistical measure that represents the proportion of the variance in the dependent variable that can be explained by the independent variables in a regression model. It ranges from 0 to 1, where 0 indicates that the independent variables have no explanatory power, and 1 indicates a perfect fit. $R^2$ is calculated as the ratio of the explained sum of squares (ESS) to the total sum of squares (TSS). RMSE is a measure of the average difference between the predicted values and the actual values in a regression or time series model. It is calculated as the square root of the mean of the squared differences between the predicted and actual values. RMSE provides a measure of the model's accuracy, with lower values indicating better performance. RMAE is similar to RMSE, but it measures the average absolute deviation between the predicted values and the actual values. It is calculated by taking the square root of the average of the absolute differences between the predicted and actual values. RMAE is another measure of the model's accuracy, with lower values indicating better performance. These statistical measures provide valuable insights into the performance and accuracy of models in various fields, such as regression analysis, time series forecasting etc.

Conceptual framework or flowchart or work flow of this study is presented in the Fig 1.

## Results

### ARIMA model fitting

While testing the stationarity of NMR, the test statistic of Kwiatkowski–Phillips–Schmidt–Shin (KPSS) test is 1.6059 with p value 0.01 which is less than 0.05. Therefore, we may reject the null hypothesis "The Data is Stationary", that means that the NMR data is nonstationary. After second differencing, KPSS test statistic is 0.090682 with p value 0.1. That means that in KPSS test, we fail to reject our null hypothesis. Therefore, NMR can be considered as stationary at 2nd differencing.

From the Fig 2, it was observed that, there was one significant spike of Autocorrelation Function (ACF) and two tentative significant spikes of Partial Autocorrelation Function (PACF).

So, considering order of p from 0 to 2, order of q from 0 to 1 and fixing the order of d at 2 all possible models were generated with $R^2$ value greater than 0.7 using R Studio and sorted them with respect to AIC shown in Table 1 with their corresponding RMAE, RMSE, $R^2$ and AIC.

In the Table 1, selecting the optimal model relies on the choice of the model with the lowest AIC. From the Table 1, it is observed that there are total six models following all the conditions. Among the models the model standing on the top of the Table 1, contains the smallest AIC value. Hence, the ARIMA (0,2,1) model is the parsimonious model for the NMR. The coefficient of the estimated ARIMA (0,2,1) model has been shown in the Table 2.

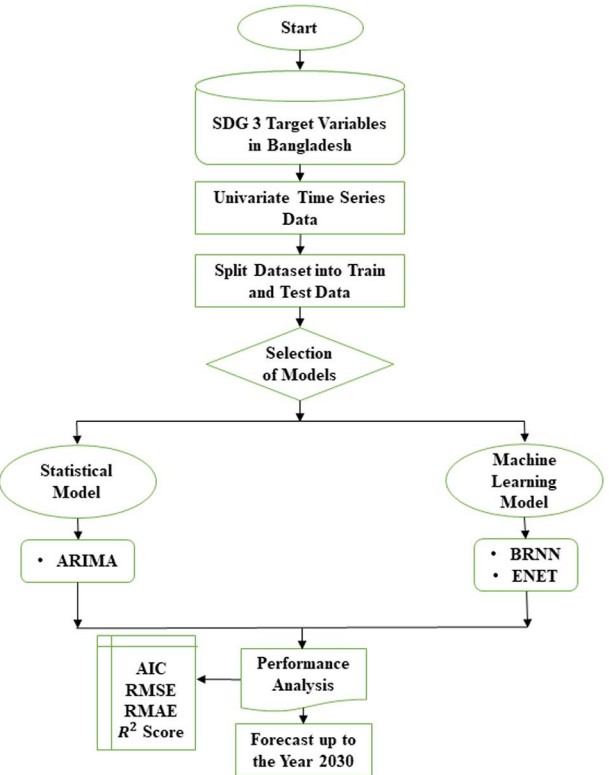

**Fig 1. Flow chart of the study.**

While testing the stationarity of U5MR, the test statistic of KPSS test is 1.5924 with p value 0.01 which is less than 0.05. Therefore, we may reject the null hypothesis "The Data is Stationary", that means that the U5MR data is nonstationary. After first differencing, KPSS test statistic is 0.0529 with p value 0.1. That means that in KPSS test, we fail to reject our null hypothesis. Therefore, NMR can be considered as stationary at first differencing.

In the Fig 3, there exits two tentative significant spikes of ACF and three tentative significant spikes of PACF.

So, considering order of p from 0 to 3, order of q from 0 to 2 and fixing the order of d at 1 all possible models were generated with $R^2$ value greater than 0.7 using R Studio and sorted them with respect to AIC are shown in Table 3 with their corresponding RMAE, RMSE, $R^2$ and AIC.

In the Table 3, selecting the optimal model relies on the choice of the model with the lowest AIC. From the Table 3, it is observed that only two models are following all the conditions. Between two of them the model standing on the top of the Table 3, contains the smallest AIC value. Hence, the ARIMA (2,1,2) model is the parsimonious model for the U5MR. Now the coefficient of the estimated ARIMA (2,1,2) model has been shown in the Table 4.

In the context of ARIMA models, a general guideline suggests that there should be a minimum of 50 observations, preferably exceeding 100, for effective model performance (Box and Tiao 1975) [29]. Since there are just 21 and 20 observations of MMR and Death Rate Due to RTI respectively, ARIMA models for those datasets are not ready to be fitted. The analyses were conducted with applying only ML methods for MMR and Death Rate Due to RTI. The model parameters for all the indicators are presented in the Table 5.

**ACF of NMR at Second Difference**

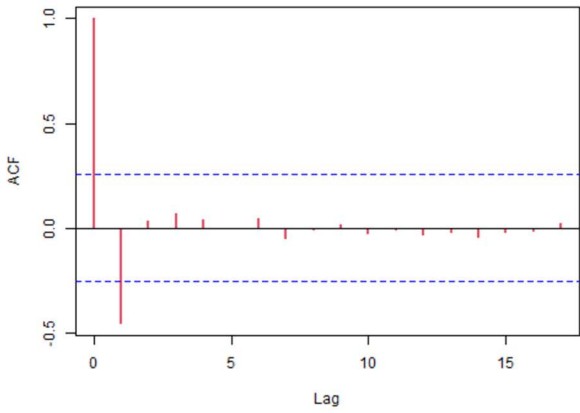

**PACF of NMR at Second Difference**

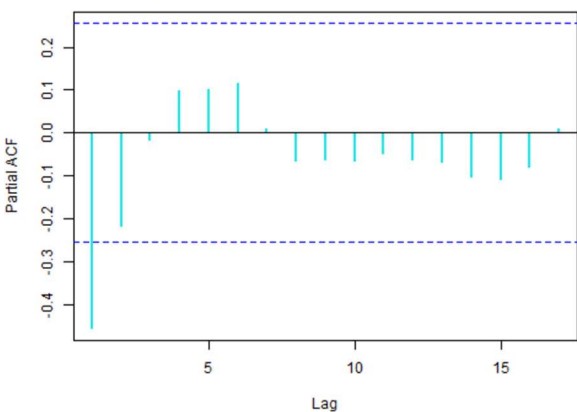

**Fig 2. The correlogram of NMR at 2nd differencing.**

**Table 1. Evaluating model adequacy using AIC for the recommended ARIMA of NMR.**

| Model | RMAE | RMSE | $R^2$ | AIC |
|---|---|---|---|---|
| **ARIMA (0,2,1)** | **1.4** | **2.3** | **0.9** | **94.6*** |
| ARIMA (2,2,0) | 1.3 | 1.9 | 0.9 | 95.5 |
| ARIMA (1,2,0) | 1.2 | 1.7 | 0.9 | 95.6 |
| ARIMA (1,2,1) | 1.3 | 2.1 | 0.9 | 95.9 |
| ARIMA (2,2,1) | 1.3 | 1.9 | 0.9 | 97.5 |
| ARIMA (0,2,0) | 1.1 | 1.4 | 0.9 | 103.7 |

## BRNN

The number of neurons in the BRNN layer in this Table 5, the values range from 2 to 18. A higher number of neurons allows the model to capture more complex patterns in the data. However, too many neurons might lead to overfitting, especially if the dataset is not large enough. The lag represents the time delay considered in the BRNN. It indicates how many past times steps the model takes into account when making predictions. A lag of 6 or 4

**Table 2. Estimated parameters for ARIMA (0,2,1).**

| Variables | Coefficients | Std. Error | t-statistic | p-value |
|---|---|---|---|---|
| MA (1) | −0.5 | 0.1 | −4.8 | < 0.001 |

$\sigma^2 = 0.3$log likelihood = −52.5

The model parameter is statistically significant because $p$ value is less than 0.05.

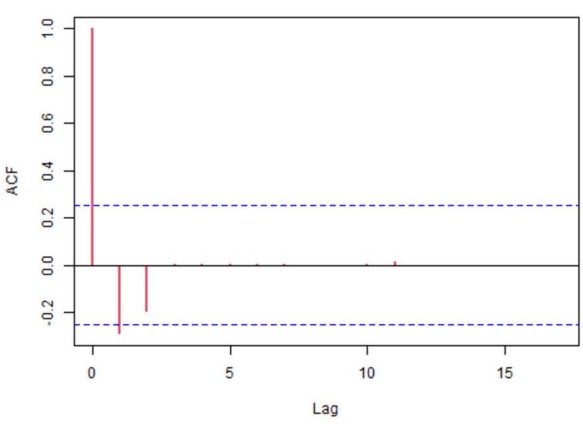

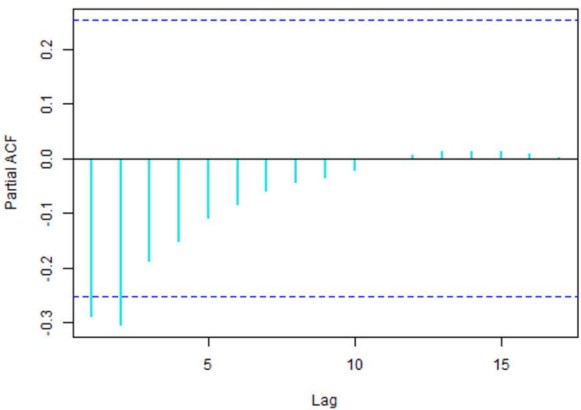

**Fig 3. The correlogram of U5MR at 1st differencing.**

**Table 3. Evaluating model adequacy using AIC for the recommended ARIMA of U5MR.**

| Model | RMAE | RMSE | $R^2$ | AIC |
|---|---|---|---|---|
| ARIMA (2,1,2) | 1.6 | 2.5 | 0.9 | 423.2* |
| ARIMA (3,1,2) | 1.3 | 2.1 | 0.9 | 425.2 |

suggests that the model considers information from the previous 6- or 4-time steps, respectively. The appropriate lag depends on the temporal dependencies in the data. The parameters vary across different health indicators (NMR, U5MR, MMR, Death Rate Due to RTI). This

**Table 4. Estimated parameters for ARIMA (2,1,2).**

| Variables | Coefficients | Std. Error | t-statistic | p-value |
|---|---|---|---|---|
| AR (1) | 1.4 | 0.1 | 11.2 | < 0.001 |
| AR (2) | −0.4 | 0.1 | −3.4 | < 0.001 |
| MA (1) | −1.9 | 0.1 | −19.0 | < 0.001 |
| MA (2) | 0.9 | 0.1 | 9.6 | < 0.001 |

$\sigma^2$ = 434.4log likelihood = −266.1

All the parameters of the model are statistically significant because their corresponding $p$ values are less than 0.05.

**Table 5. Parameters of ML algorithms.**

| | BRNN | | ENET | | |
|---|---|---|---|---|---|
| Variables | Neuron | Lag | Fraction | Lambda | Lag |
| **NMR** | 18 | 6 | 0.5 | 7.40E-05 | 20 |
| **U5MR** | 2 | 6 | 0.6 | 0.000277 | 20 |
| **MMR** | 17 | 4 | 0.7 | 2.90E-05 | 4 |
| **Death Rate Due to RTI** | 17 | 2 | 0.9 | 2.27E-05 | 2 |

The model parameters are gained through the random search technique with a tunelength of 1000.

variability reflects the need to adapt the models to the specific characteristics of each mortality rate. Different mortality rates (NMR, U5MR, MMR, Death Rate Due to RTI) have different sets of parameters, indicating that the model is adapted to the specific characteristics of each variable. This is a common practice to tailor models to the nuances of different datasets. The lag parameter (6 for NMR, 6 for U5MR, 4 for MMR, and 2 for Death Rate Due to RTI) suggests that the model takes into account information from a certain number of past time steps. The appropriate lag depends on the temporal dependencies in the data. The choice of the number of neurons in the BRNN layer reflects a trade-off between model complexity and the ability to capture patterns in the data. Too few neurons might result in an oversimplified model, while too many neurons might lead to overfitting. Understanding the significance of these values is crucial for making informed decisions about the model's architecture and its ability to effectively capture patterns and relationships in the data.

## ENET

The "fraction" parameter in the Table 5 is the mixing parameter ($\alpha$) in the context of the ENET algorithm. The mixing parameter controls the balance between the $L1$ (Lasso) and $L2$ (Ridge) regularization terms. It determines the relative weight given to the $L1$ and $L2$ regularization terms. When $\alpha$ is 0, it corresponds to Ridge regression, and when $\alpha$ is 1, it corresponds to Lasso regression. A value of $\alpha$ between 0 and 1 indicates a mix of both $L1$ and $L2$ regularization. Lambda is the regularization parameter in the ENET algorithm. It controls the strength of the penalty applied to the model coefficients. A smaller lambda value implies weaker regularization, allowing the model to fit the training data more closely. Regularization helps prevent overfitting by penalizing overly complex models. The lag parameter is specified as 20 for both NMR and U5MR, and 4 and 2 for MMR and Death Rate Due to RTI respectively. In the context of ENET, this lag value is related to the structure of the ENET algorithm in handling multicollinearity and performing variable selection. The ENET parameters (alpha, lambda, lag) appear to be consistent across different health indicators (NMR, U5MR,

MMR, Death Rate Due to RTI). This suggests a standardized approach in applying the ENET algorithm to various mortality rate variables. The choice of the mixing parameter (α) reflects the balance between feature selection (Lasso) and feature grouping (Ridge). A higher α tends to result in sparser models, potentially selecting a subset of important features. The choice of lambda values suggests a trade-off between fitting the data closely and preventing overfitting. A careful selection of lambda is essential to find the right balance between model complexity and generalization.

The performances of the selected models are presented in the Fig 4.

From the Fig 4, it seems that ENET is performing better than the other models. Performance measures of different models for all the variables are given below.

The Table 6 presents two performance metrics RMAE and RMSE, for three different models ENET, BRNN, and ARIMA across four health indicators: NMR, U5MR, MMR, and Death Rate Due to RTI. The lower the value of these criteria, the better the model's forecasting accuracy. The ENET model demonstrates superior performance across the indicators, with lower RMAE and RMSE values compared to BRNN and ARIMA. For instance, using cross-validation procedures, in the case of NMR, ENET achieves an RMAE of 0.603446 and an RMSE of 0.451162,

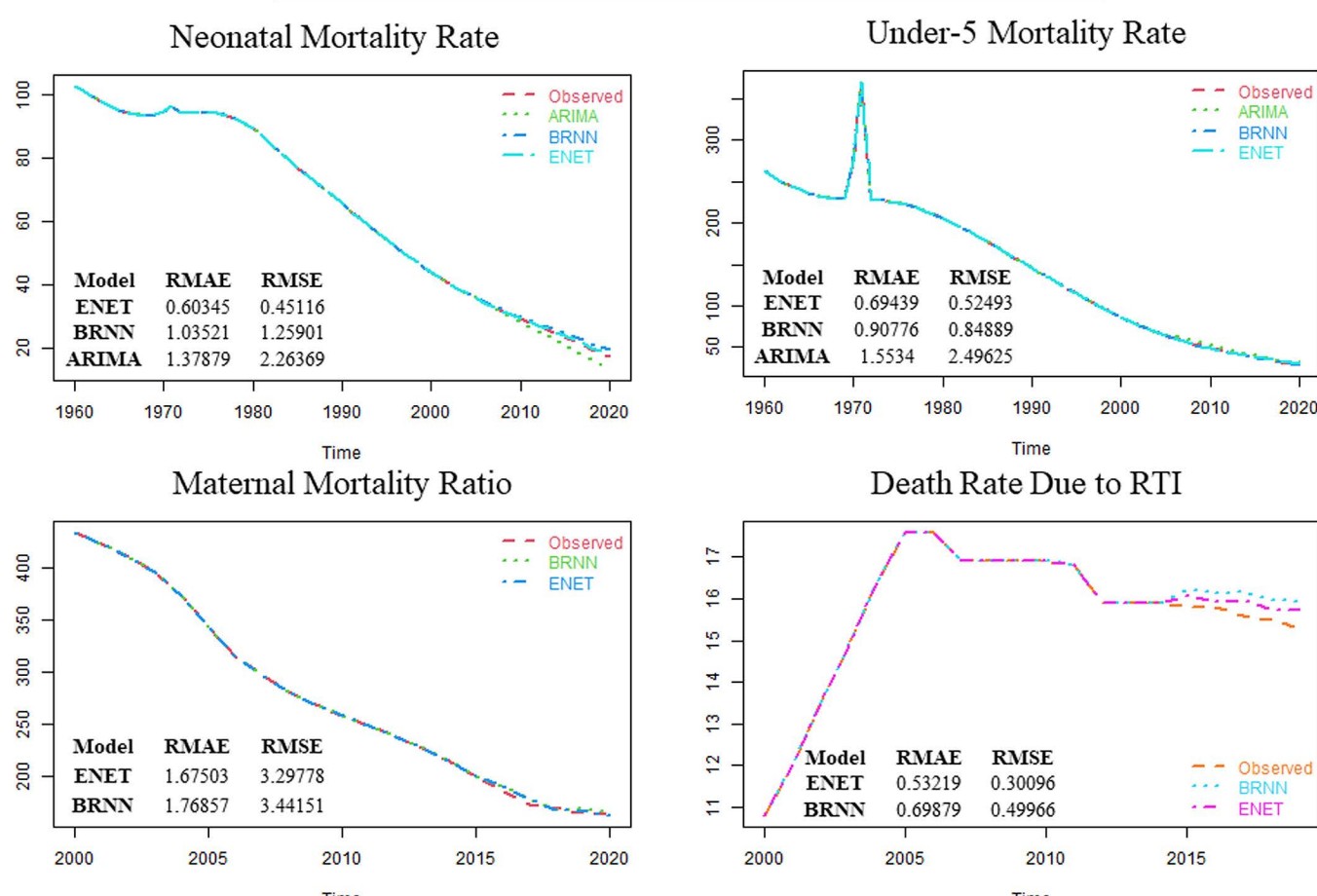

**Fig 4. Performance measure of all the selected models.**

outperforming both BRNN and ARIMA. Similarly, for U5MR, MMR, and Death Rate Due to RTI, ENET consistently exhibits lower error metrics compared to the other models. However, it is notable that ARIMA was not applied for MMR and Death Rate Due to RTI. Overall, the findings suggest that the ENET model may offer enhanced accuracy in predicting these critical health indicators compared to alternative models such as BRNN and ARIMA. The performance rankings of these three models for four target variables are shown in Table 7.

In Bangladesh, the machine-learning model ENET is recommended as better model than BRNN and ARIMA for forecasting NMR, U5MR, MMR and Death Rate Due to RTI. Appropriate models on NMR, U5MR, MMR and Death Rate Due to RTI are fitted and the forecasted results for all the selected models are presented in the Table 8.

**Table 6. Performance measures of different models.**

| Series | NMR | | U5MR | | MMR | | Death Rate Due to RTI | |
|---|---|---|---|---|---|---|---|---|
| Model | RMAE | RMSE | RMAE | RMSE | RMAE | RMSE | RMAE | RMSE |
| ENET | 0.603446 | 0.451162 | 0.694388 | 0.524926 | 1.675027 | 3.297782 | 0.532194 | 0.300964 |
| BRNN | 1.035213 | 1.259006 | 0.90776 | 0.848894 | 1.768568 | 3.441505 | 0.698786 | 0.499657 |
| ARIMA | 1.378789 | 2.263687 | 1.5534 | 2.496248 | – | – | – | – |

**Table 7. Ranking of the models as per forecasting accuracy.**

| Variables | BRNN | ENET | ARIMA |
|---|---|---|---|
| NMR | 2 | 1 | 3 |
| U5MR | 2 | 1 | 3 |
| MMR | 2 | 1 | – |
| Death Rate Due to RTI | 2 | 1 | – |

**Table 8. Forecasting results for all selected models.**

| Year | Forecasting Results of NMR | | | | Forecasting Results of U5MR | | | | Forecasting Results of MMR | | | Forecasting Results of Death Rate Due to RTI | | |
|---|---|---|---|---|---|---|---|---|---|---|---|---|---|---|
| | Obs. | ARIMA | BRNN | ENET | Obs. | ARIMA | BRNN | ENET | Obs. | BRNN | ENET | Obs. | BRNN | ENET |
| 2015 | 23.6 | 23.6 | 23.5 | 23.5 | 38.1 | 37.1 | 37.9 | 38.1 | 200 | 202.1 | 203.3 | 15.8 | 15.8 | 15.8 |
| 2016 | 22.4 | 22.4 | 22.4 | 22.3 | 36.1 | 35.0 | 36.0 | 36.1 | 186 | 188.9 | 190.3 | 15.8 | 15.9 | 15.9 |
| 2017 | 20.9 | 21.2 | 21.2 | 20.9 | 34.2 | 33.0 | 34.2 | 34.2 | 173 | 176.8 | 177.6 | 15.6 | 15.9 | 15.9 |
| 2018 | 19.7 | 19.5 | 19.9 | 19.7 | 32.4 | 31.2 | 32.6 | 32.4 | 169 | 165.8 | 166.0 | 15.5 | 15.7 | 15.8 |
| 2019 | 18.5 | 18.4 | 18.7 | 18.5 | 30.7 | 29.6 | 31.1 | 30.7 | 165 | 166.3 | 164.3 | 15.3 | 15.7 | 15.7 |
| 2020 | 17.5 | 17.2 | 17.6 | 17.4 | 29.1 | 28.1 | 29.7 | 29.1 | 163 | 163.4 | 161.1 | – | 15.6 | 15.6 |
| 2021 | – | 16.3 | 16.5 | 16.4 | – | 26.8 | 28.4 | 27.5 | – | 158.9 | 158.2 | – | 15.8 | 15.8 |
| 2022 | – | 15.2 | 15.5 | 15.4 | – | 25.2 | 27.3 | 26.0 | – | 154.6 | 153.3 | – | 16.0 | 16.0 |
| 2023 | – | 14.1 | 14.6 | 14.5 | – | 23.9 | 26.2 | 24.5 | – | 149.5 | 148.3 | – | 16.2 | 16.2 |
| 2024 | – | 13.1 | 13.9 | 13.6 | – | 22.9 | 25.3 | 23.1 | – | 145.4 | 144.1 | – | 16.3 | 16.3 |
| 2025 | – | 11.9 | 13.2 | 12.7 | – | 21.9 | 24.4 | 21.7 | – | 142.3 | 140.4 | – | 16.4 | 16.3 |
| 2026 | – | 10.8 | 12.6 | 11.8 | – | 21.1 | 23.6 | 20.4 | – | 140.3 | 137.2 | – | 16.4 | 16.3 |
| 2027 | – | 9.7 | 12.1 | 10.8 | – | 20.3 | 22.9 | 19.1 | – | 138.8 | 134.2 | – | 16.3 | 16.2 |
| 2028 | – | 8.6 | 11.7 | 9.9 | – | 19.5 | 22.3 | 17.9 | – | 137.4 | 131.5 | – | 16.2 | 16.2 |
| 2029 | – | 7.5 | 11.4 | 9.0 | – | 18.8 | 21.7 | 16.7 | – | 135.9 | 128.9 | – | 16.2 | 16.1 |
| **2030** | – | **6.4** | **11.0** | **8.1** | – | **18.2** | **21.2** | **15.5** | – | **134.3** | **126.6** | – | **16.1** | **16.1** |

Obs. stands for observed.

From the Table 8, the forecasted values of NMR in 2030 for ARIMA, BRNN and ENET models are 6.4, 11.0 and 8.1 respectively. The forecasted values of U5MR in 2030 for ARIMA, BRNN and ENET models are 18.2, 21.2 and 15.5 respectively. The forecasted values of MMR in 2030 for BRNN and ENET models are 134.3 and 126.6 respectively. The forecasted values of Death Rate Due to RTI in 2030 for both BRNN and ENET models is 16.1. Here, comparisons among the forecasted values using different models of the indicators with their corresponding SDG 3 target for the year 2030 are presented in the Table 9.

From Table 9, all the forecasted values of all selected models are less than the target value for NMR and U5MR which are 12 and 25 respectively. On the other hand, all the forecasted values of the models are greater than the target value for MMR and Death Rate Due to RTI which are 70 and 1.2 respectively.

## Discussion

The research presented a comprehensive analysis of SDG 3 in the context of Bangladesh. The study employed both time series models, specifically ARIMA, and ML algorithms, including BRNN and ENET, to forecast these indicators up to the year 2030. The findings indicate a positive trajectory for Bangladesh in achieving its 2030 targets for NMR and U5MR. The forecasted values consistently fall below the respective targets, suggesting significant progress in reducing NMR and U5MR. This optimistic outlook aligns with the country's historical achievements in these areas, forecasted from the parsimonious model as the decline in NMR to 8.1 in 2030 and the decrease in U5MR to 15.5 deaths per 1,000 live births during the same period. However, challenges persist in meeting the 2030 targets for MMR and Death Rate Due to RTI. The forecasted values for these indicators surpass the set targets, indicating a need for intensified efforts to address maternal mortality and RTI. The research emphasizes the significance of SDG 3 in Bangladesh's journey toward sustainable development in health and well-being. The combination of time series models and ML algorithms provides a nuanced understanding of future trends in health indicators. The adoption of ENET as the recommended ML model highlights its superiority in forecasting NMR, U5MR, MMR, and Death Rate Due to RTI compared to other models. The study's contribution lies in its holistic approach, considering multiple indicators and utilizing advanced forecasting techniques. The analysis not only celebrates Bangladesh's progress in reducing child mortality but also calls attention to areas requiring targeted interventions. The research serves as a valuable resource for policymakers, guiding them in allocating resources and devising strategies to address the remaining challenges in maternal health and road safety. While Bangladesh demonstrates promising strides in achieving SDG 3 targets for child mortality, concerted efforts are essential to overcome challenges related to maternal health and RTI.

**Table 9. Presentation of SDG 3 targets with corresponding forecasted values.**

| Variables | Target for Year 2030 | Forecasted Values | | |
|---|---|---|---|---|
| | | ARIMA | BRNN | ENET |
| **NMR** | 12 | 6.4 | 11.0 | 8.1 |
| **U5MR** | 25 | 18.2 | 21.2 | 15.5 |
| **MMR** | 70 | – | 134.3 | 126.6 |
| **Death Rate Due to RTI** | 1.2 | – | 16.1 | 16.1 |

## Conclusions

After model fitting and forecasting, drawing the conclusion on the overall findings is necessary. When comparing all the parsimonious classical and ML models in terms of RMAE and RMSE, the analysis indicates that ENET is considered as better model for NMR, U5MR, MMR, and Death Rate Due to RTI, followed by BRNN and ARIMA (Note: ARIMA was not applied for MMR and Death Rate Due to RTI). ENET generally outperforms BRNN and ARIMA, exhibiting lower RMAE and RMSE values for indicators such as NMR, U5MR, and Death Rate Due to RTI. For instance, in the case of NMR, ENET achieves an RMAE of 0.603446 and an RMSE of 0.451162, outperforming both BRNN and ARIMA. Similarly, for U5MR, MMR, and Death Rate Due to RTI, ENET consistently exhibits lower error metrics compared to the other models. Overall, the results suggest that ENET may offer superior predictive accuracy for these health indicators compared to BRNN and ARIMA models. The value of NMR for ENET to the year 2030 will be 8.1 for Bangladesh, falling below the pre-assigned target value for the indicator to the year 2030. And all of the forecasted values are less than the pre assigned target value for the indicator which is 12 by the year 2030. So, the target of NMR is going to be fulfilled. The value of U5MR for ENET to the year 2030 will be 15.5, falling below the pre-assigned target value for the indicator, which is 25 by the year 2030. So, the target of U5MR is also going to be fulfilled. The forecasted values of MMR for BRNN and ENET to the year 2030 are 134.3 and 126.6 respectively, all exceeding the pre-assigned target value for the indicator, which is 70 by the year 2030. So, MMR is going to fail to meet the target of 2030. The forecasted values of Death Rate Due to RTI for both BRNN and ENET to the year 2030 is 16.1, surpassing the pre-assigned target value for the indicator, which is 1.2 by the year 2030. So, Death Rate Due to RTI is also going to fail to meet the target of 2030. Since the study is providing strong evidence against the fulfilment of the SDG target for the indicators MMR (SDG Indicator 3.1.1) and Death Rate Due to RTI (SDG Indicator 3.6.1) by the year 2030, the policymakers must need to take necessary steps to overcome this situation of those two indicators. This research extends the existing body of knowledge on SDG 3 targets and health indicators in Bangladesh. By employing a wide array of modelling techniques, including advanced ML methods, this study provides a holistic assessment of progress and challenges in achieving the Sustainable Development Goals related to health. In the future, there is great potential for using other ML methods to forecast health indicators and assess progress towards SDGs in Bangladesh. Deep learning models, gradient boosting models, time series forecasting with transformers, Bayesian methods, and ensemble methods are some ML approaches that can be explored. These methods offer opportunities to improve predictive accuracy, capture complex relationships, handle uncertainty, and enhance overall forecasting performance. Rigorous experimentation and evaluation, considering data availability, computational resources, and interpretability, will be crucial for selecting and implementing appropriate ML methods in this context.

## Supporting information

**S1 Data.  Exploring the achievements and forecasting of SDG 3 using machine learning algorithms: Bangladesh perspective.**
(XLSX)

## Author contributions

**Conceptualization:** Md. Maeen Molla, Md. Ayub Ali, Mst. Papia Sultana, Dulal Chandra Roy.

**Data curation:** Md. Maeen Molla, Md. Raqibul Islam.

**Formal analysis:** Md. Maeen Molla, Md. Sifat Hossain, Md. Raqibul Islam.

**Investigation:** Mst. Papia Sultana.

**Methodology:** Md. Maeen Molla, Md. Ayub Ali, Mst. Papia Sultana.

**Software:** Md. Maeen Molla, Md. Sifat Hossain.

**Supervision:** Dulal Chandra Roy.

**Validation:** Md. Maeen Molla, Md. Ayub Ali, Dulal Chandra Roy.

**Visualization:** Md. Maeen Molla, Md. Sifat Hossain, Md. Raqibul Islam.

**Writing – original draft:** Md. Maeen Molla, Md. Sifat Hossain.

**Writing – review & editing:** Md. Maeen Molla, Dulal Chandra Roy.

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
