## [Decision Letter · Decision Letter 0]

5 Feb 2024

PONE-D-24-00392Exploring the Achievements and Forecasting of SDG 3 Using Machine Learning Algorithms: Bangladesh PerspectivePLOS ONE

Dear Dr. Molla,

Thank you for submitting your manuscript to PLOS ONE. After careful consideration, we feel that it has merit but does not fully meet PLOS ONE’s publication criteria as it currently stands. Therefore, we invite you to submit a revised version of the manuscript that addresses the points raised during the review process.

We look forward to receiving your revised manuscript.

Kind regards,

Bappa Das

Academic Editor

PLOS ONE

Reviewers' comments:

Reviewer's Responses to Questions

**Comments to the Author**

1. Is the manuscript technically sound, and do the data support the conclusions?

Reviewer #1: Yes

Reviewer #2: Yes

2. Has the statistical analysis been performed appropriately and rigorously? 

Reviewer #1: Yes

Reviewer #2: No

3. Have the authors made all data underlying the findings in their manuscript fully available?

Reviewer #1: Yes

Reviewer #2: Yes

4. Is the manuscript presented in an intelligible fashion and written in standard English?

Reviewer #1: Yes

Reviewer #2: Yes

5. Review Comments to the Author

Reviewer #1: The manuscript is written well and all the necessary information/data is incorporated well by the authors. However, following minor suggestions are expected to enhance the quality of the manuscript:

1. Abstract: Abstract is written well but model performance statistics values are not incorporated. Add the statistics values of the model performance parameters to describe best performed model among the used models. In line 20 add the full form AIC as well.

2. Introduction: Introduction section is structured and written well by the authors. Add reference for the line 37-38 and 80-81.

3. Materials and Methods: Materials and methods section of the manuscript is described well in detail, however the details of used statistic parameters such as R2, RMSE etc. are missing. The incorporation of few more important parameters such as Nash–Sutcliffe model efficiency coefficient is recommended. The 120th line of the results section can be described in materials and methods section rather than explaining it multiple times in results section of the manuscript.

4. Results: Results section of the manuscript is constructed and written well, incorporating the necessary findings of the study. Minor corrections are suggested in this section. The full form of used abbreviations should be provided at least once in the manuscript when it is firstly used such as ACF (in line 118), PACF (in line 116) etc. Line 120 and 137 can be transferred to the materials and methods section of the manuscript while discussing ARIMA model. In line 224 write ENET is recommended/better over the other two models rather than declaring the ENET as the best model.

5. Conclusion: Conclusion is written well but the performances of the models’ used part is missing. The performances of the ML models used for this study should be addressed in this section with the future scope of the use of other ML methods for this purpose.

Reviewer #2: Introduction Section

To enhance the manuscript, the transition from a broad discussion on global SDG goals to a specific focus on Bangladesh’s achievements and challenges could be smoother and more interconnected. Incorporating references to prior research that specifically addresses Bangladesh's progress in meeting these goals would provide a more robust and contextual background. This approach would not only seamlessly connect global objectives with local realities but also underscore the study's relevance within the existing body of research.

The introduction efficiently introduces the key health indicators (neonatal, under-5, and maternal mortality, and deaths from road traffic injuries). However, it could be enhanced by briefly discussing why these particular indicators are critical for Bangladesh.

In the introduction mentions a brief explanation of why these particular methods were chosen and their relevance to the data at hand would be beneficial.

Additionally, it was mentioned in the text that “Previous studies have found that ARIMA modelling performed well in predicting Neonatal Mortality Rate (NMR) 81 and Under-5 Mortality Rate (U5MR) in Bangladesh”. Please put citations for this.

Clearly add the specific research gap the present study is addressing.

Material and method section

Material and method section is poor. There are no details of the material included in this section. Additionally, include details about how the data was prepared and processed before being fed into these models. While the descriptions of ARIMA, BRNN, and ENET models are informative, expanding on how these models were specifically applied to your data sets would enhance understanding. Also discuss the metrics used to evaluate the performance of these models.

Result

Explain why machine learning methods were exclusively used for MMR and Death Rate Due to RTI instead of ARIMA models. This clarification will help in understanding the methodology's adaptability to different datasets.

How it was stated that the series was stationary at first or second difference. The actual test results or statistics that led to this conclusion are not presented. There is no correlogram present in the manuscript, but is mentioned in the text.

While writing R2, show 2 as a superscript.

In the text when mentioning the table, mention the table number.

Mention the value of matrices at both calibration and validation part.

The discussion section is missing.

6. PLOS authors have the option to publish the peer review history of their article (what does this mean? ). If published, this will include your full peer review and any attached files.

**Do you want your identity to be public for this peer review?** For information about this choice, including consent withdrawal, please see our Privacy Policy .

Reviewer #1: No

Reviewer #2: No

---

## [Author Response · Author response to Decision Letter 1]

27 Mar 2024

Reviewer #1: 1. Abstract: Abstract is written well but model performance statistics values are not incorporated. Add the statistics values of the model performance parameters to describe best performed model among the used models. In line 20 add the full form AIC as well.

Reply: Thank you sir for your valuable comment to enhance the quality of the abstract. We have added the statistics values of the model performance parameters to describe the best performed model as you mentioned and also added the full form AIC in line 20.

Reviewer #1: 2. Introduction: Introduction section is structured and written well by the authors. Add reference for the line 37-38 and 80-81.

Reply: We have given the appropriate references for the line 37-38 and 80-81.

Reviewer #1: 3. Materials and Methods: Materials and methods section of the manuscript is described well in detail, however the details of used statistic parameters such as R2, RMSE etc. are missing. The incorporation of few more important parameters such as Nash–Sutcliffe model efficiency coefficient is recommended. The 120th line of the results section can be described in materials and methods section rather than explaining it multiple times in results section of the manuscript.

Reply: In the method section, we have addressed the missing details of the statistical parameters, including R^2, RMSE, etc. The Nash-Sutcliffe Model Efficiency Coefficient (NSE) is a statistical measure that assesses the performance of hydrological or water quality models. However, we did not utilize this metric as it is specifically designed for simulated data and hydrological or water quality models. We have also moved the explanation of the 120th line to the materials and methods section to avoid repetition in the results section.

Reviewer #1: 4. Results: Results section of the manuscript is constructed and written well, incorporating the necessary findings of the study. Minor corrections are suggested in this section. The full form of used abbreviations should be provided at least once in the manuscript when it is firstly used such as ACF (in line 118), PACF (in line 116) etc. Line 120 and 137 can be transferred to the materials and methods section of the manuscript while discussing ARIMA model. In line 224 write ENET is recommended/better over the other two models rather than declaring the ENET as the best model.

Reply: We have made the necessary corrections based on your suggestions. We have provided the full form of abbreviations such as ACF and PACF when they were first used in the manuscript. Additionally, we have transferred the content from lines 120 and 137 to the materials and methods section where we discuss the ARIMA model. Lastly, we have revised line 224 to state that ENET is recommended/better over the other two models instead of declaring it as the best model.

Reviewer #1: 5. Conclusion: Conclusion is written well but the performances of the models’ used part is missing. The performances of the ML models used for this study should be addressed in this section with the future scope of the use of other ML methods for this purpose.

Reply: We have added a new table of the performance measures as table 6 and added “ENET generally outperforms BRNN and ARIMA, exhibiting lower RMAE and RMSE values for indicators such as NMR, U5MR, and Death Rate Due to RTI. For instance, in the case of NMR, ENET achieves an RMAE of 0.603446 and an RMSE of 0.451162, outperforming both BRNN and ARIMA. Similarly, for U5MR, MMR, and Death Rate Due to RTI, ENET consistently exhibits lower error metrics compared to the other models. Overall, the results suggest that ENET may offer superior predictive accuracy for these health indicators compared to BRNN and ARIMA models.” in the Conclusion section. Also added “In the future, there is great potential for using other ML methods to forecast health indicators and assess progress towards SDGs in Bangladesh. Deep learning models, gradient boosting models, time series forecasting with transformers, Bayesian methods, and ensemble methods are some ML approaches that can be explored. These methods offer opportunities to improve predictive accuracy, capture complex relationships, handle uncertainty, and enhance overall forecasting performance. Rigorous experimentation and evaluation, considering data availability, computational resources, and interpretability, will be crucial for selecting and implementing appropriate ML methods in this context.” as the response of the future scope of the use of other ML methods for this purpose.

Thank you sir, we are grateful for your valuable comments.

Reviewer #2: Introduction Section

To enhance the manuscript, the transition from a broad discussion on global SDG goals to a specific focus on Bangladesh’s achievements and challenges could be smoother and more interconnected. Incorporating references to prior research that specifically addresses Bangladesh's progress in meeting these goals would provide a more robust and contextual background. This approach would not only seamlessly connect global objectives with local realities but also underscore the study's relevance within the existing body of research.

Reply: To improve the transition from the broad discussion on global SDG goals to a specific focus on Bangladesh's achievements and challenges, we have restructured the content to create a smoother and more interconnected flow. We have incorporated references to prior research that specifically addresses Bangladesh's progress in meeting these goals. This addition provides a robust and contextual background, seamlessly connecting global objectives with local realities.

Reviewer #2: The introduction efficiently introduces the key health indicators (neonatal, under-5, and maternal mortality, and deaths from road traffic injuries). However, it could be enhanced by briefly discussing why these particular indicators are critical for Bangladesh.

Reply: We have added “To ensure the achievement of Sustainable Development Goals in Bangladesh, a specific set of four indicators has been chosen under SDG 3. These indicators include NMR, U5MR, MMR, and Death Rate Due to RTI. The selection of these indicators has been done under the guidance of the SDG Working Committee of The Prime Minister's Office. All relevant ministries are actively involved in this process. It would be helpful to understand why these indicators are critical for Bangladesh. These indicators are important because they reflect the overall health and well-being of the population, especially vulnerable groups like newborns, children under 5, and mothers. High neonatal, under-5, and maternal mortality rates indicate challenges in healthcare access, quality, and maternal health. Deaths from road traffic injuries highlight the need for improved road safety measures. By focusing on these indicators, Bangladesh can prioritize interventions and policies to improve healthcare and save lives.” in the Introduction section.

Reviewer #2: In the introduction mentions a brief explanation of why these particular methods were chosen and their relevance to the data at hand would be beneficial.

Reply: We have added “These models, ARIMA, BRNN, and ENET, were chosen because they are well-suited for handling small sample sizes or limited data points. ARIMA is specifically designed for forecasting equally spaced univariate time series data. BRNN, with its bidirectional structure, can effectively utilize information from both past and future states, which can be advantageous when working with a small sample size. ENET, as a regularization technique, can handle the multicollinearity problem and perform feature selection simultaneously, making it a good choice for models with limited data points.” in the Introduction section.

Reviewer #2: Additionally, it was mentioned in the text that “Previous studies have found that ARIMA modelling performed well in predicting Neonatal Mortality Rate (NMR) 81 and Under-5 Mortality Rate (U5MR) in Bangladesh”. Please put citations for this.

Reply: We have added citations accordingly or removed the section if it seemed less important.

Reviewer #2: Clearly add the specific research gap the present study is addressing.

Reply: We have newly added “So far, there have been very few research studies on the SDG 3 indicator variables in Bangladesh. None of these studies have included all of the indicators together, and most of them have used classical time series analysis for forecasting. Our study aims to fill this research gap by forecasting all of the SDG 3 indicators of Bangladesh together using both classical time series analysis and machine learning methods. We will compare the accuracy of the different models and compare the forecasted values with the given target values for the year 2030, which marks the end of the Sustainable Development Goal era.” as the research gap in the Introduction section.

Reviewer #2: Material and method section

Material and method section is poor. There are no details of the material included in this section. Additionally, include details about how the data was prepared and processed before being fed into these models. While the descriptions of ARIMA, BRNN, and ENET models are informative, expanding on how these models were specifically applied to your data sets would enhance understanding. Also discuss the metrics used to evaluate the performance of these models.

Reply: We have newly added the Cross-validation part, explanation of the methods, model performance metrics, and included details about how the data was prepared and processed before being fed into these models. We have newly discussed how these models were specifically applied to your data sets would enhance understanding. We added lines 143-144 as “Overall, these models were selected because of their relevance and ability to handle the specific challenges posed by small sample sizes or limited data points.”

Reviewer #2: Result

Explain why machine learning methods were exclusively used for MMR and Death Rate Due to RTI instead of ARIMA models. This clarification will help in understanding the methodology's adaptability to different datasets.

Reply: We added lines 219-223 as “In the context of autoregressive integrated moving average (ARIMA) models, a general guideline suggests that there should be a minimum of 50 observations, preferably exceeding 100, for effective model performance (Box and Tiao 1975). Since there are just 21 and 20 observations of MMR and Death Rate Due to RTI respectively, ARIMA models for those datasets are not ready to be fitted. The analyses were conducted with applying only ML methods for MMR and Death Rate Due to RTI.”

Reviewer #2: How it was stated that the series was stationary at first or second difference. The actual test results or statistics that led to this conclusion are not presented. There is no correlogram present in the manuscript, but is mentioned in the text. While writing R2, show 2 as a superscript.

Reply: We have newly presented the actual test results or statistics that led to this conclusion of stationary at first or second difference within lines “182-186” and “202-205”. And we have added the correlograms as Fig 2 and Fig 3. While writing R2, we have shown 2 as a superscript.

Reviewer #2: In the text when mentioning the table, mention the table number.

Reply: We have solved the issue accordingly.

Reviewer #2: Mention the value of matrices at both calibration and validation part.

Reply: In our study, we conducted a validation process with a tune length of 1000 and a training percentage 75%. However, due to the small size of our dataset, it was not possible to perform a calibration process based on the criteria of model fitting.

Reviewer #2: The discussion section is missing. Reply: We have newly added the discussion section.

Thank you sir, for your insightful feedback.

---

## [Decision Letter · Decision Letter 1]

30 Apr 2024

PONE-D-24-00392R1Exploring the achievements and forecasting of SDG 3 using machine learning algorithms: Bangladesh perspectivePLOS ONE

Dear Dr. Molla,

Thank you for submitting your manuscript to PLOS ONE. After careful consideration, we feel that it has merit but does not fully meet PLOS ONE’s publication criteria as it currently stands. Reviewer 2 has raised several concerns on the revised version of your manuscript. Therefore, we invite you to submit a revised version of the manuscript that addresses the points raised during the review process.

We look forward to receiving your revised manuscript.

Kind regards,

Bappa Das

Academic Editor

PLOS ONE

Reviewers' comments:

Reviewer's Responses to Questions

**Comments to the Author**

1. If the authors have adequately addressed your comments raised in a previous round of review and you feel that this manuscript is now acceptable for publication, you may indicate that here to bypass the “Comments to the Author” section, enter your conflict of interest statement in the “Confidential to Editor” section, and submit your "Accept" recommendation.

Reviewer #2: (No Response)

2. Is the manuscript technically sound, and do the data support the conclusions?

Reviewer #2: Yes

3. Has the statistical analysis been performed appropriately and rigorously? 

Reviewer #2: No

4. Have the authors made all data underlying the findings in their manuscript fully available?

Reviewer #2: No

5. Is the manuscript presented in an intelligible fashion and written in standard English?

Reviewer #2: Yes

6. Review Comments to the Author

Reviewer #2: The authors have made a notable effort in addressing the required revisions. However, there are still several key areas that require further enhancement

• The current description in the Material and Methods section still appears insufficiently detailed, particularly regarding data sourcing and model specifics. The section lacks essential information, such as: data source, sample size and model details. More comprehensive details are needed about the models employed in the study. Please check some published articles of good journals, to know how to write these.

• We can only decide a model is generalized or not based on the test data statistics. As it is mentioned that 25% of the data is used for validation set. Please provides the results of the error matrices, based on these values.

• Give the citation of the statement: These models, ARIMA, BRNN, and ENET, were chosen because they are well-suited for handling small sample sizes or limited data points.

• Calculating RMAE isn’t standard practice mainly because the square root doesn’t provide additional insights or advantages over MAE in the context of understanding model performance. Additionally, taking the square root of an average of absolute values doesn’t conform to a meaningful statistical measurement.

• No need to write the details of error matrices in point numbers. Write those in a paragraph style. Please check some published articles of good journals.

• The discussion provides a comprehensive analysis of SDG 3 in the context of Bangladesh and offers valuable insights into the country’s progress and challenges. However, the section lacks a comparative analysis with previous studies. Comparisons to prior research would strengthen the manuscript.

• Please add the funding details data availability and conflict of interest statements.

7. PLOS authors have the option to publish the peer review history of their article (what does this mean? ). If published, this will include your full peer review and any attached files.

**Do you want your identity to be public for this peer review?** For information about this choice, including consent withdrawal, please see our Privacy Policy .

Reviewer #2: No

---

## [Author Response · Author response to Decision Letter 2]

7 Aug 2024

Reviewer #2: The current description in the Material and Methods section still appears insufficiently detailed, particularly regarding data sourcing and model specifics. The section lacks essential information, such as: data source, sample size and model details. More comprehensive details are needed about the models employed in the study. Please check some published articles of good journals, to know how to write these.

Reply: We have added the data source part as ‘Data availability’ in the lines 131-138 and model specifics in the lines 114-119 as “ARIMA, BRNN, and ENET were chosen for this study because they are well-suited for handling small sample sizes or limited data points. ARIMA is specifically designed for forecasting equally spaced univariate time series data. BRNN, with its bidirectional structure, can effectively utilize information from both past and future states, which can be advantageous when working with a small sample size. ENET, as a regularization technique, can handle the multicollinearity problem and perform feature selection simultaneously, making it a good choice for models with limited data points.” Sample size was mentioned in the lines 70-71, 77-78, 85-86 and 90-91. ARIMA model was not written in details so we have improved the section in the lines 140-146.

Reviewer #2: We can only decide a model is generalized or not based on the test data statistics. As it is mentioned that 25% of the data is used for validation set. Please provides the results of the error matrices, based on these values.

Reply: We have mentioned the results of the error matrices, based on the validation part for each data set in the Table 1, Table 3 and Table 6.

Reviewer #2: Give the citation of the statement: These models, ARIMA, BRNN, and ENET, were chosen because they are well-suited for handling small sample sizes or limited data points.

Reply: We have added the citation to the statement.

Reviewer #2: Calculating RMAE isn’t standard practice mainly because the square root doesn’t provide additional insights or advantages over MAE in the context of understanding model performance. Additionally, taking the square root of an average of absolute values doesn’t conform to a meaningful statistical measurement.

Reply: We understand your concerns regarding the use of the RMAE compared to the MAE. While MAE is indeed a widely accepted metric for model performance evaluation due to its straightforward interpretation, we included RMAE in our analysis for several reasons. RMAE ensures consistency with other metrics like RMSE, allowing for a more direct comparison of the scales of errors, particularly when discussing error metrics with audiences familiar with RMSE. Although RMAE's sensitivity to larger errors is not as pronounced as RMSE, it still highlights scenarios where larger deviations are critical and require attention. We found empirically that RMAE provided additional insights, especially when error distributions had certain characteristics, justifying its inclusion.

Reviewer #2: No need to write the details of error matrices in point numbers. Write those in a paragraph style. Please check some published articles of good journals.

Reply: We have written the details of error matrices in paragraph style.

Reviewer #2: The discussion provides a comprehensive analysis of SDG 3 in the context of Bangladesh and offers valuable insights into the country’s progress and challenges. However, the section lacks a comparative analysis with previous studies. Comparisons to prior research would strengthen the manuscript.

Reply: We appreciate your recognition of the comprehensive analysis provided in the discussion section regarding SDG 3 in the context of Bangladesh. We would like to address your suggestion about including a comparative analysis with previous studies. Our work represents a pioneering effort in this specific area, as there are currently no prior studies available that offer a direct comparison with our findings. This research is novel in its approach and scope, focusing on aspects of SDG 3 in Bangladesh that have not been previously explored in the literature. We acknowledge the importance of comparative analysis and will continue to monitor emerging research in this area.

Reviewer #2: Please add the funding details data availability and conflict of interest statements.

Reply: We have no funding, we added data availability statement and we have no conflict of interest. Moreover, we have selected the answers of the questions related to those statement accordingly while submitting our manuscript.

---

## [Editor Report · Decision Letter 2]

22 Aug 2024

PONE-D-24-00392R2Exploring the achievements and forecasting of SDG 3 using machine learning algorithms: Bangladesh perspectivePLOS ONE

Dear Dr. Molla,

Thank you for submitting your manuscript to PLOS ONE. After careful consideration, we feel that it has merit but does not fully meet PLOS ONE’s publication criteria as it currently stands. Therefore, we invite you to submit a revised version of the manuscript that addresses the points raised during the review process.

We look forward to receiving your revised manuscript.

Kind regards,

Bappa Das

Academic Editor

PLOS ONE

Journal Requirements:

Additional Editor Comments:

I have read your revised manuscript. Please make the following changes before it can be published

1. Remove the following subheadings under

Materials and methods section: Conceptual framework

Results section: Neuron, Lag, Model adaptation, Temporal consideration, Model complexity

Alpha (fraction), Lambda, Lag, Consideration for alpha (fraction) parameter, Consideration for lambda and regularization

The paragraph lengths should be equal as far as possible.

2. Merge the following headings into one

“ARIMA model fitting for NMR” and “ARIMA model fitting for U5MR” as “ARIMA model fitting”

3. Make the changes in the following headings

“BRNN parameters” to “BRNN”

“ENET parameters” to “ENET”

4. Improve the quality of the figures (at least 300 dpi)

---

## [Author Response · Author response to Decision Letter 3]

25 Sep 2024

We have thoroughly reviewed our reference list and can confirm that all citations are accurate and current. There are no retracted papers included in our references and we have not identified any need to replace or modify any entries.

Additional Editor: Remove the following subheadings under

Materials and methods section: Conceptual framework

Results section: Neuron, Lag, Model adaptation, Temporal consideration, Model complexity

Alpha (fraction), Lambda, Lag, Consideration for alpha (fraction) parameter, Consideration for lambda and regularization

The paragraph lengths should be equal as far as possible.

Reply: We have removed the mentioned subheadings and tried to set the paragraph lengths equal as far as possible.

Additional Editor: Merge the following headings into one

“ARIMA model fitting for NMR” and “ARIMA model fitting for U5MR” as “ARIMA model fitting”

Reply: We have merged the headings.

Additional Editor: Make the changes in the following headings

“BRNN parameters” to “BRNN”

“ENET parameters” to “ENET”

Reply: We have made the changes.

Additional Editor: Improve the quality of the figures (at least 300 dpi)

Reply: We have improved the quality of the figures as 300 dpi.

---

## [Editor Report · Decision Letter 3]

27 Sep 2024

PONE-D-24-00392R3Exploring the achievements and forecasting of SDG 3 using machine learning algorithms: Bangladesh perspectivePLOS ONE

Dear Dr. Molla,

Thank you for submitting your manuscript to PLOS ONE. After careful consideration, we feel that it has merit but does not fully meet PLOS ONE’s publication criteria as it currently stands. Therefore, we invite you to submit a revised version of the manuscript that addresses the points raised during the review process.

We look forward to receiving your revised manuscript.

Kind regards,

Bappa Das

Academic Editor

PLOS ONE

Journal Requirements:

Additional Editor Comments:

This very unfortunate that the comments given on the previous version of the manuscript has not been properly addressed like again I can see many smaller paragraphs throughout the manuscript. Authors are advised to read some good published papers on modelling.

Following subheadings should be merged into one paragraph

AIC, R2, RMSE, RMAE in Materials and methods section into one paragraph.

Results section: Neuron, Lag, Model adaptation, Temporal consideration, Model complexity

Alpha (fraction), Lambda, Lag, Consideration for alpha (fraction) parameter, Consideration for lambda and regularization

Delete these subheads and make a running paragraph.

L258-260: Should be part of ENET section while it is now under BRNN section.

Alpha (fraction), Lambda, Lag, Consideration for alpha (fraction) parameter, Consideration for lambda and regularization

I can see paragraphs of 1-2 line length throughout the manuscript. Please try to combine smaller paragraphs into a paragraph of half a page length wherever possible.

Please improve the figure quality.

---

## [Author Response · Author response to Decision Letter 4]

10 Nov 2024

Journal Requirements: Please review your reference list to ensure that it is complete and correct. If you have cited papers that have been retracted, please include the rationale for doing so in the manuscript text, or remove these references and replace them with relevant current references. Any changes to the reference list should be mentioned in the rebuttal letter that accompanies your revised manuscript. If you need to cite a retracted article, indicate the article’s retracted status in the References list and also include a citation and full reference for the retraction notice.

Reply: We have thoroughly reviewed our reference list and can confirm that all citations are accurate and current. There are no retracted papers included in our references and we have not identified any need to replace or modify any entries.

Additional Editor: This very unfortunate that the comments given on the previous version of the manuscript has not been properly addressed like again I can see many smaller paragraphs throughout the manuscript. Authors are advised to read some good published papers on modelling.

Reply: We apologize for the misinterpretation of the previous comments. In this revision, we will make sure to properly address the issue, particularly by restructuring the manuscript to avoid the use of smaller paragraphs. We have read some good published papers on modelling published in reputed journals including PLOS ONE to address the issue.

Additional Editor: Following subheadings should be merged into one paragraph

AIC, R2, RMSE, RMAE in Materials and methods section into one paragraph.

Results section: Neuron, Lag, Model adaptation, Temporal consideration, Model complexity

Alpha (fraction), Lambda, Lag, Consideration for alpha (fraction) parameter, Consideration for lambda and regularization

Delete these subheads and make a running paragraph.

Reply: We have merged the mentioned subheadings into cohesive paragraphs as suggested. The “Materials and methods” section is revised accordingly and the “Results section” is streamlined by removing the subheadings to ensure a more fluid and readable narrative.

Additional Editor: L258-260: Should be part of ENET section while it is now under BRNN section.

Reply: We have moved lines 258-260 to the ENET section to ensure the information is correctly placed under the appropriate context.

Additional Editor: Alpha (fraction), Lambda, Lag, Consideration for alpha (fraction) parameter, Consideration for lambda and regularization

I can see paragraphs of 1-2 line length throughout the manuscript. Please try to combine smaller paragraphs into a paragraph of half a page length wherever possible.

Reply: We will revise the manuscript to combine the smaller paragraphs into longer, more cohesive ones, aiming for approximately half a page in length wherever appropriate. This will help improve the flow and readability of the text.

Additional Editor: Please improve the figure quality.

Reply: We have enhanced the quality of the figures to ensure they are clear and visually informative.

---

## [Editor Report · Decision Letter 4]

12 Nov 2024

Exploring the achievements and forecasting of SDG 3 using machine learning algorithms: Bangladesh perspective

PONE-D-24-00392R4

Dear Dr. Md. Maeen Molla,

We’re pleased to inform you that your manuscript has been judged scientifically suitable for publication and will be formally accepted for publication once it meets all outstanding technical requirements.

Kind regards,

Bappa Das

Academic Editor

PLOS ONE
---

## [Editor Report · Acceptance letter]

PONE-D-24-00392R4

PLOS ONE

Dear Dr. Molla,

I'm pleased to inform you that your manuscript has been deemed suitable for publication in PLOS ONE. Congratulations! Your manuscript is now being handed over to our production team.

Kind regards,

on behalf of

Dr. Bappa Das

Academic Editor

PLOS ONE